# Managing FAIR Tribological Data Using Kadi4Mat

**Nico Brandt** [1,*], **Nikolay T. Garabedian** [1,2], **Ephraim Schoof** [3], **Paul J. Schreiber** [1,2], **Philipp Zschumme** [1], **Christian Greiner** [1,2] and **Michael Selzer** [1,4]

1   Institute for Applied Materials (IAM-CMS), Karlsruhe Institute of Technology (KIT), Straße am Forum 7, 76131 Karlsruhe, Germany; nikolay.garabedian@kit.edu (N.T.G.); paul.schreiber@cellcentric.net (P.J.S.); philipp.zschumme@kit.edu (P.Z.); christian.greiner@kit.edu (C.G.); michael.selzer@kit.edu (M.S.)
2   KIT IAM-CMS MicroTribology Center (μTC), Straße am Forum 5, 76131 Karlsruhe, Germany
3   Helmholtz Institute Ulm for Electrochemical Energy Storage (HIU), Helmholtzstraße 11, 89081 Ulm, Germany; ephraim.schoof@kit.edu
4   Institute for Digital Materials Science (IDM), Karlsruhe University of Applied Sciences, Moltkestraße 30, 76133 Karlsruhe, Germany
*   Correspondence: nico.brandt@kit.edu

**Abstract:** The ever-increasing amount of data generated from experiments and simulations in engineering sciences is relying more and more on data science applications to generate new knowledge. Comprehensive metadata descriptions and a suitable research data infrastructure are essential prerequisites for these tasks. Experimental tribology, in particular, presents some unique challenges in this regard due to the interdisciplinary nature of the field and the lack of existing standards. In this work, we demonstrate the versatility of the open source research data infrastructure Kadi4Mat by managing and producing FAIR tribological data. As a showcase example, a tribological experiment is conducted by an experimental group with a focus on comprehensiveness. The result is a FAIR data package containing all produced data as well as machine- and user-readable metadata. The close collaboration between tribologists and software developers shows a practical bottom-up approach and how such infrastructures are an essential part of our FAIR digital future.

**Keywords:** research data management; FAIR data; digitisation; open source; materials science; tribology

## 1. Introduction

Data-intensive applications, such as machine learning, data mining, or more generally, data science, are becoming increasingly important in all areas of research to generate new knowledge due to the growing amount of data obtained from experiments and simulations [1,2]. These areas also include engineering sciences and, in particular, the broad field of materials science, where the full potential of digitisation is far from exhausted [3]. While various data science methods are already used in the preparation, characterisation, and production of materials, one aspect is usually neglected: the management and analysis of data [4]. Most of the generated research data are only used and stored locally, while published data often suffer from problems such as non-standardised formats, the lack of rich metadata and the fact that "negative" results tend to not get published at all [5]. The FAIR data principles [6] offer guidelines to tackle these challenges, requiring that research data are findable, accessible, interoperable and reusable. These guidelines have also been adopted by funding agencies such as the German Research Foundation [7] and have led to the development of detailed metrics assessing the FAIRness of research data and corresponding metadata [8].

To meet these requirements, proper and consistent research data management is essential; this is not trivial to implement and usually requires a research data infrastructure as a technical foundation. Common components of research data infrastructures include electronic laboratory notebooks (ELNs) for capturing data and—often unstructured—metadata close to the source, research data repositories for publishing the data, and everything in

between. Such systems not only enable more powerful data analyses, but also aid in comparisons between theoretical and experimental data, provide reproducible workflows, and allow other researchers to reference and reuse the data. Typically, research data software can either be discipline-specific or generic. For the former, there are repositories such as the Materials Project [9] or the NOMAD Repository [10]. However, since materials science in particular is a very heterogeneous and interdisciplinary field of research [11], there are not many such specialised systems. To the authors' best knowledge, there are currently no specific ELNs that directly serve a broad range of the field. Examples of generic systems include data repositories such as Zenodo [12], ELNs such as eLabFTW [13], and virtual research environments such as the Galaxy Project [14]. Since all of the above systems are open source, they can usually be further customised and often installed on institutional servers, which may also be appropriate for materials science workflows. However, the additional work required for end users or system administrators to customise the system to their specific needs can be a significant hurdle, as smaller workgroups in particular lack the necessary resources and technical knowledge.

One sub-discipline of materials science in which the above points are particularly evident is tribology—a highly interdisciplinary field that itself presents some unique challenges. In tribology, there are no universal ways for quantifying processes such as friction and wear, laboratory equipment is often custom-built, and tribological experiments are often adjusted "on the fly". Thus, there is a lack of discipline-specific research data infrastructure to support the recording of the entire sequence of events and external influences in tribological workflows, which makes the generation of FAIR data very difficult. Although the need to digitise such experiments has been discussed for some time [15,16], little progress has been made recently. Previous efforts have focused mainly on providing web-based databases with mostly numerical properties of individual tribological datasets, such as the Tribocollect [17] database, which is still available at the time of writing. However, access to the full database is fee-based, and it is not clear whether the database is still actively developed and maintained. Publication of tribological data, either alone or as a supplement to a text publication, is also not common in tribology, with publications such as [18] being one of the only examples. Even then, these data usually lack the extensive and, most importantly, machine-readable metadata needed to actually reproduce the published results, which already fails to meet the last point of the FAIR guidelines, namely data reusability.

Based on the above challenges, it is clear that data exchange between tribologists is not possible without a complete semantic description of the data describing their properties. While metadata schemas exist for higher-level disciplines, such as for computational engineering [19] or materials science [20], the defined terms are usually too general to adequately capture the specifics of typical tribological workflows. In addition to the classic metadata describing each piece of data, the relationships between data must also be captured to trace the processed data back to their origin for complete data provenance. A common method for implementing such requirements are ontologies, which can help organise data and metadata based on formal definitions, properties, and relationships that are restricted to a specific domain. Since creating a suitable ontology from scratch is a difficult task, many existing ontologies build upon higher- or intermediate-level ontologies. Such an approach is also taken by the tribAIn [21] ontology, which currently appears to be the only existing ontology for tribology. This ontology aims to formalise all kinds of tribological knowledge that is mainly contained in text-based publications using natural language, which requires a high level of manual annotation and is ultimately limited to content that authors have decided to include. However, to take full advantage of an ontology, the generation of metadata should be as close to the source as possible while conducting an experiment to minimise human error. An appropriate research data infrastructure must provide capabilities to represent such types of metadata and relationships as defined in ontologies while being flexible enough to handle heterogeneous data formats and experimental workflows. The technical infrastructure can thus be seen as a bridge

between the controlled terminology as defined by the experimenters and the data being produced. In order to meet all these requirements, both the definition of terminologies and ontologies as well as the development of the necessary technical infrastructure should be done simultaneously in close collaboration between experimenters and software developers. While such an approach can usually lead to overspecialisation of the infrastructure to a specific area, the interdisciplinary nature of tribology requires generic approaches that are easily transferable to other research areas.

This paper shows how the research data infrastructure Kadi4Mat [22,23] was used and further developed to achieve the above tasks. As a showcase and basis for the developments, a relatively standard tribological experiment—lubricated pin-on-disc—is conducted by a workgroup at the Institute for Applied Materials (IAM-CMS) of the Karlsruhe Institute of Technology (KIT). The setup of such an experiment consists of a lubrication bath and a stationary pin (the counter body) that is pressed against a rotating disc (the base body) under a certain load with the help of a tribometer's load cell in order to measure the resulting friction and wear. Since the experiment is kept comparatively simple, the focus can be on the data generation and management aspects, while the experimental output remains manageable. In addition, the tribologists have no prior experience with the research data infrastructure Kadi4Mat or similar software. The goal of this collaboration is to harness Kadi4Mat's potential to produce and publish what is called a *FAIR data package* [24] containing all raw and processed data with the appropriate metadata in both human-readable and machine-readable formats. Since the actual showcase experiment, including data and metadata preparation, is performed by the tribologists, this paper focuses mainly on the technical aspects related to the use and development of Kadi4Mat in the context of creating this data package based on the generic functionality described in [22]. This includes some additional tools that have been developed to further bridge the gap between the existing technical solutions and the specific requirements of the experimenters. All other aspects, as well as the FAIR data package itself, are described in more detail in a sister publication [25], which aims to provide a blueprint for FAIR data generation and publication in experimental tribology.

## 2. Materials and Methods

Kadi4Mat (Karlsruhe Data Infrastructure for Materials Science) is an open source research data infrastructure being developed at the IAM-CMS of the KIT. Although the software has been developed with materials science in mind, it has been kept as generic as possible. This aspect is a direct result of the heterogeneity of the materials science disciplines and also allows a much easier adaptation of the infrastructure for other domains. The software is best described as a virtual research environment that combines features of ELNs and repositories. A central aspect of the ELN component is the automated and documented execution of heterogeneous workflows, which is currently under development [22]. Workflows are a generic concept that describes a well-defined sequence of sequential or parallel steps that are processed as automatically as possible, prompting for user input as needed. In practice, this may involve running an analysis tool on a computer or controlling and retrieving data from an experimental device. While this puts the focus on automation, some typical features of classic ELNs are still implemented, such as the inclusion of handwritten notes or sketches. The flexible design of Kadi4Mat thus allows events to be recorded without a planned workflow, making it a comprehensive meeting point for experimenters and modellers. In contrast to the ELN, the repository component represents the structured data management and primarily has the storage of the actual data and metadata as its goal. In general, the focus is on warm data, i.e., unpublished data that need further analysis. Despite the logical division into the two components, they act as a single system from a user and implementation perspective.

Kadi4Mat, like many systems of this type, is web-based and offers a corresponding graphical user interface (GUI). Thus, even inexperienced users can utilise all functions of Kadi4Mat with the help of a regular, modern web browser and easily collaborate with

other researchers without having to install additional software. Since a classic client-server architecture is employed, system administrators can deploy Kadi4Mat locally or centrally for individual workgroups or entire institutions. For advanced and automated use, a REST-like [26] HTTP application programming interface (API) is available that allows access to most of Kadi4Mat's functionality by sending appropriate HTTP requests to the various endpoints of the API. Because JSON is used as the exchange format, the API provides standardised, machine-readable representations of all resources. There is also an open source wrapper library written in Python—called kadi-apy [27]—that provides both an object-oriented approach to work with the API and a command line interface (CLI) for integration with simple scripts and other programming languages. In addition to using the API on an individual level, offering a standardised and easy-to-use API is particularly important for integrating existing systems.

The central component of Kadi4Mat are the so-called records, which combine data with corresponding metadata and support time-stamped, uniquely identifiable and traceable revision capabilities to track changes to these metadata. A record can represent any type of digital or physical object, including raw and processed data, experimental samples, or even devices that would typically only consist of metadata. In addition to basic, fixed metadata such as the title, description or type of a record, each record can contain an arbitrary amount of generic metadata that can be specified in the form of a structured, JSON-like composition which provides a high degree of flexibility. As with most features, specifying this type of metadata is possible either via the API or the GUI of Kadi4Mat. To facilitate the manual entry of metadata when using the latter, templates can be created. These are useful not only in cases where existing metadata schemas are to be applied to a multitude of records, but can also help to facilitate the establishment of new metadata standards where none yet exist, even if they are initially only used by individual users. To specify relationships between multiple records, they can be linked together in the form of triples. Each triple represents a unidirectional link that contains both records and the metadata of the link itself, currently including a type and timestamp. In this way, relationships such as "'experiment' 'uses' 'device'" can be represented, where the device can be a metadata-only record that can be linked to any experiment using that device. Once created, records can be exported in various ways. In addition to machine-readable formats such as JSON, user-readable formats such as PDF or graphical visualisations of the record relationships are available. Furthermore, it is possible to publish the contents of a record directly to an external repository via the GUI of Kadi4Mat, which is currently implemented for the Zenodo repository. This functionality is integrated via a plugin interface, which means that the required authentication and subsequent publishing process can be implemented separately from the main code base. To improve the organisation of records, multiple records can be aggregated into one or more possibly nested collections that represent logical groupings, for example in the form of projects, sub-projects and experiments, with the latter containing the actual experimental inputs, outputs and processes as records. Each resource can be shared individually with other users or groups of users using role-based access permissions [22].

While this paper focuses on the technical aspects of FAIR data production using Kadi4Mat, a lot of preliminary work has been done by the tribologists, as shown in Figure 1, as the first part of the data production pipeline.

In addition to conducting the actual tribological experiment, suitable terms were collected to describe all steps and processes of the experiment. This was done as a collaborative effort within the experimental workgroup using a locally hosted instance of MediaWiki [28]. Since such systems are mostly used for unstructured metadata, which are often unsuitable for further processing, the software Protégé [29] was used to develop an ontology (called TriboDataFAIR [30]) based on these terms in order to transform them into a structured and controlled vocabulary. Together with Kadi4Mat and some complementary technical solutions, this ontology forms the basis for the digitisation of experimental processes and outputs.

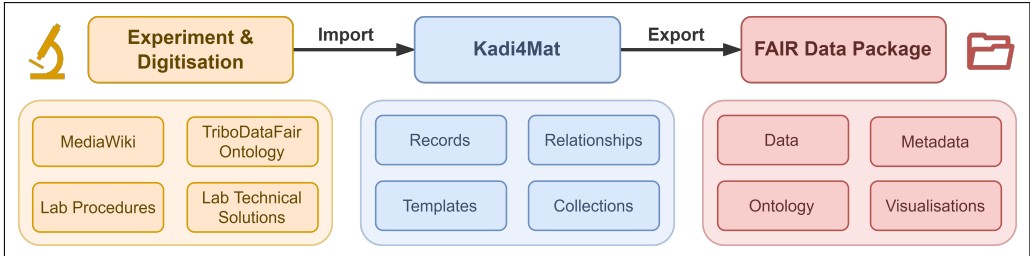

**Figure 1.** Visualisation of the data production pipeline, starting with the showcase experiment and its digitisation and ending with the complete FAIR data package. Kadi4Mat acts as an intermediary between both steps by enabling structured data and metadata storage and providing the necessary import and export functionality.

## 3. Results

In order to support the tribologists and identify their specific needs, an instance of Kadi4Mat first had to be made available. Although it is possible to set up a local installation, an existing centralised instance hosted at the IAM-CMS was used. This not only eliminates the installation and administration work for the tribologists, but also facilitates collaboration with external researchers outside the workgroup. For the collaboration in the workgroup itself, a group with all tribologists was created within Kadi4Mat, which was granted appropriate roles to be able to view and edit all created resources, while external collaborators were assigned individual roles with varying permissions.

As shown in Figure 1, Kadi4Mat acts mainly as an intermediary between the showcase experiment, the structured management of the resulting data and the bundling of the FAIR data package for publication. In this case, from the perspective of the research data infrastructure, the pipeline begins with the import of all necessary data and metadata in the form of records, which was done in both "analogue" (manual) and "digital" (automatic) form. Since many processes in experimental tribology are inherently analogue, i.e., there is no automatic way to bridge the gap between an events' occurrence and the accounting thereof via digital data and metadata (e.g., wiping a solvent by hand), most of the records in this project were created manually using the GUI provided by Kadi4Mat. However, some of the metadata are always captured automatically, such as the creator of each record and its creation date. All records were grouped into a single collection representing the entire showcase experiment. An example of such a record, as displayed via the GUI, is shown in Figure 2.

This record represents one of the main experimental samples, namely the test-ready counter body used in the showcase experiment, which was produced by performing various preparation steps on the raw material. Figure 3 shows this sample in the context of the entire experimental workflow.

Since this record is the digital counterpart of a physical object, it consists only of metadata, which are typically entered manually for this kind of record. The actual experimental entity that makes up the record is mostly described by the generic metadata, so the manual creation of this type of metadata in particular needs to be facilitated. Kadi4Mat provides a powerful editor to efficiently enter such metadata, which at their core consist of potentially nested key-value pairs of different types, some of which are shown in Figure 2.

The aforementioned template functionality is one of the main features that further streamlines this process. Currently, templates can be considered mainly as blueprints that provide a common structure and metadata for entire records or only their generic metadata, although both types of templates can also be combined. For generic metadata in particular, templates allow basic validation instructions to be specified, such as required values or a list of predefined values to select from when applying the template. Like records, templates can be created using both the GUI or the API of Kadi4Mat, either directly or by exporting the metadata of an existing record as a new template. Similar to the metadata, most templates used by the tribologists were created manually.

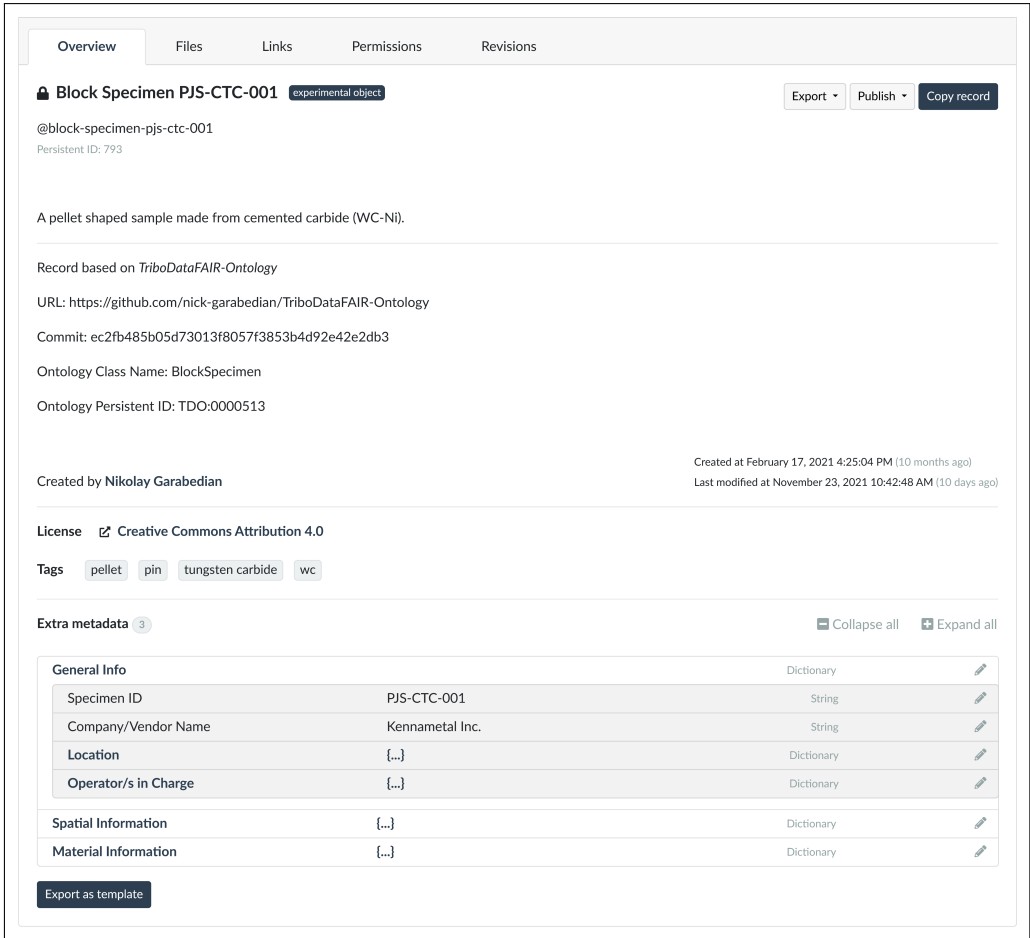

**Figure 2.** Screenshot showing an overview of a record in Kadi4Mat representing one of the main experimental samples, namely the test-ready counter body/pin of the pin-on-disc setup used in the showcase experiment. The basic metadata of the record, including title, description, user information and tags, as well as the (condensed) generic metadata of the example (indicated by *extra metadata*) are shown. The menu shown at the top can be used to navigate to different views to edit the record or view its files, relationships and permissions.

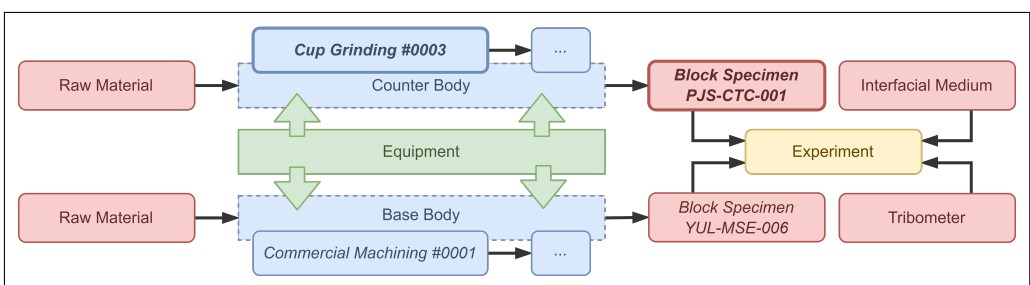

**Figure 3.** Simplified timeline of the objects and processes involved in the showcase experiment, starting with the raw materials and ending with the actual experiment involving the test-ready, pre-processed samples—the counter body (*Block Specimen PJS-CTC-001*, as also shown in Figure 2) and base body (*Block Specimen YUL-MSE-006*)—the interfacial medium (lubricant) and the tribometer. Each individual preparation process (e.g., *Cup Grinding #0003*) and sample is represented as a record. A more detailed version of this timeline can be found in the sister publication [25].

For both the metadata and the templates, the TriboDataFAIR ontology was used as a basis. Since the ontology was created using the Web Ontology Language (OWL) [31], a tool called SurfTheOWL [32] was developed to restructure the information contained in the ontology from a class hierarchy to a "metadata hierarchy" that corresponds to the

generic metadata structure expected by Kadi4Mat. The core of this tool is based on the Python library owlready2 [33], while the Python web framework Django [34] was used to implement a graphical, web-based representation of the metadata structure that serves as the basis for manual metadata entry and template creation. Compared to the ontology itself, this representation is easy enough to understand to allow non-experts to verify the consistency of the metadata and templates with the information contained in the ontology. The structure can also be exported in a machine-readable JSON format, which can be used as a basis for automating template creation. For ease of use, all of SurfTheOWL's code can be bundled and exported as a stand-alone executable file to reach people who do not have the software otherwise required to run the tool.

While the use of templates helps to reduce errors in manual metadata entry, automatic collection of metadata at the source should be the preferred method whenever possible. In addition to reducing human error to a minimum, well-maintained automated data acquisition tools can eliminate a lot of manual work, allowing researchers to focus on their actual experiments. Furthermore, all types of experimental data and metadata are always collected and annotated, including negative results, which may prove to be equally important. The tribometers used by the tribologists are usually controlled with LabView [35], which allows the execution and parametrisation of any commands installed on the same machine as LabView, for example by using PowerShell if running on Windows. The CLI of the kadi-apy wrapper library, which was implemented specifically to support the integration of programming languages outside of Python, was used in this context to communicate with Kadi4Mat's HTTP API. Since the kadi-apy library is compatible with Windows, the appropriate command to create a new record was added to the end of an existing experimental LabView workflow. The relevant data and metadata created as part of this existing workflow, representing the configuration and outputs of using the tribometer, were then attached to the newly created record using additional commands. For the experimental metadata, some further processing was required to convert it into the JSON structure required by Kadi4Mat, which was implemented directly in LabView. A small section of the appended workflow is shown in Figure 4, which shows the record creation and some of the common parametrisation of all kadi-apy commands that were used.

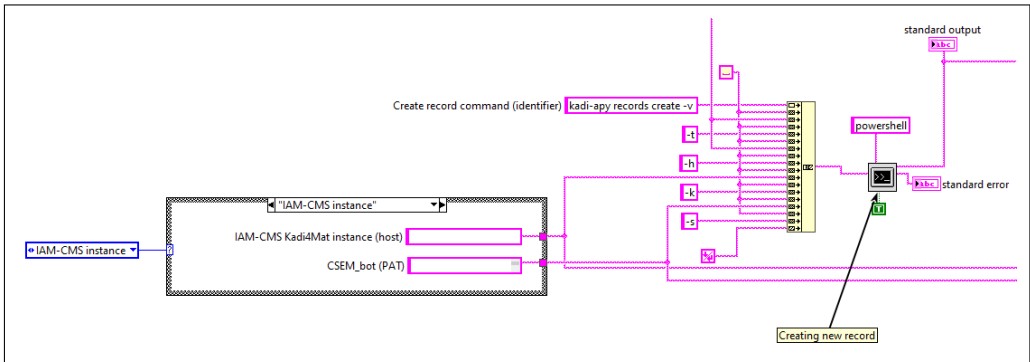

**Figure 4.** Section of a LabView workflow showing on the right a command provided by the kadi-apy library to create a new record (*kadi-apy records create*), executed via Windows PowerShell. For simplicity, the command is parametrised directly in LabView, including the host of the Kadi4Mat instance to be used and a corresponding personal access token, as shown in the box on the left. In subsequent commands, additional data and metadata are added to the new record in a similar way.

In order to authenticate with Kadi4Mat's API, a personal access token must first be created via the GUI of Kadi4Mat. Each token is bound to and managed by the respective user who creates it. Typically, a separate token with a defined lifetime is created for each application or use case, which can additionally be restricted to certain scopes or resources for security purposes. The tokens can then be managed by the kadi-apy library by storing them in a configuration file alongside the respective host of the Kadi4Mat instance to be used.

While the previous sections describe how the records themselves were created using Kadi4Mat, specifying the relationships between them is equally important. As mentioned earlier, this can be achieved using record links, which represent a unidirectional relationship between two records. Each link contains a timestamp as well as a name describing the type of relationship, which is currently freely definable. Similar to the metadata, all relationships between the different samples, devices and processes of the showcase experiment were derived from the TriboDataFAIR ontology with the help of SurfTheOWL and created manually via the GUI of Kadi4Mat. For example, the two highlighted records shown in Figure 3—*Cup Grinding #0003* and *Block Specimen PJS-CTC-001*—were linked with the relationship *physicallyModifies*, as the process represented by the former record physically alters the counter body to ensure an even bottom surface. This relationship can also be seen in Figure 5, which shows a visualisation of all incoming and outgoing links of a given record, in this case the record representing the test-ready counter body, which can be generated directly via Kadi4Mat's GUI.

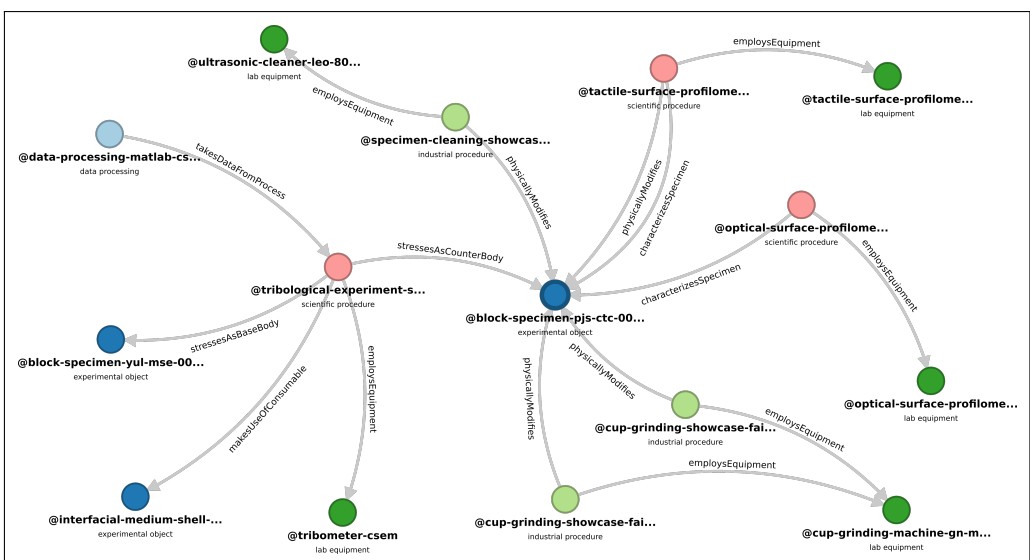

**Figure 5.** Visualisation of record links created with Kadi4Mat, starting with the record represented by the blue node in the centre. This record represents the test-ready counter body used in the showcase experiment, as also shown in Figure 2 and highlighted in Figure 3. The colour of each node denotes the type of the underlying record, for example, the light green nodes represent industrial processes that physically modify the counter body. All nodes that are up to two edges away from the start record are shown.

This type of visualisation provides a basic, ontology-like graph that can be further customised by manually dragging the nodes representing each record or by exporting the graph as an SVG image.

Since visualising the logical connections of the records does not provide an actual timeline of the experimental steps, a different kind of visualisation was created via a small Python script [36] using the kadi-apy library and Graphviz [37], as shown in Figure 6.

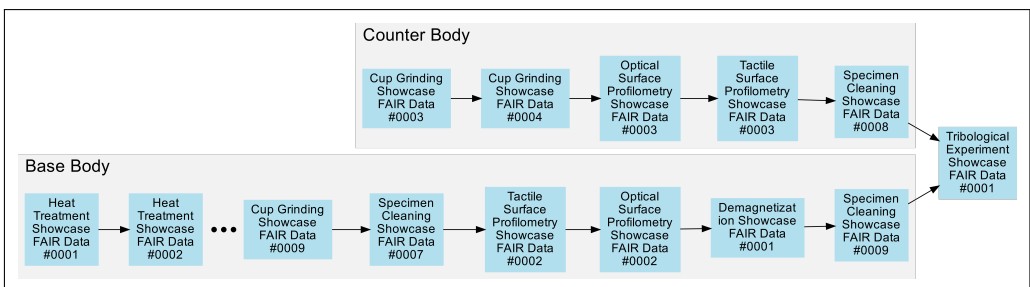

**Figure 6.** Automatically generated and condensed visualisation of the main tribological samples (counter and base body) and the processes associated with their preparation created by retrieving the incoming links of the two sample records using the kadi-apy library and Graphviz. Each sample is displayed in a separate timeline with all associated processes, ordered by the timestamp of each corresponding record. The timestamps indicate the time when the respective process was performed, which is stored as part of the generic record metadata, rather than the creation date of the individual records.

The visualisation focuses on the processes involved in preparing the two most important tribological samples, the counter and the base body. Because the focus is on the order of the processes rather than their relationships, the resulting timeline is much more similar to the manually created one shown in Figure 3. While a certain amount of code must be written for such a task, custom visualisations can easily be created in this way and, if written somewhat generically, these can easily be adapted for similar use cases. In addition, the automated visualisation was used to verify the consistency of the record links and the metadata of the corresponding records.

Once the records are created with all their data, metadata and links, there are several ways to retrieve them. While Kadi4Mat's API offers the most flexibility in this regard and can also be used for tasks other than querying existing data, there are several ways to bundle and export all of a record's metadata in various formats via both the API and the GUI. These currently include JSON, PDF and QR codes. While the API already allows retrieval of JSON data, the JSON export makes it easier for users unfamiliar with the API to retrieve all record metadata in a machine-readable format that is commonly used and understood. The JSON export also bundles record information that would normally have to be retrieved through multiple API endpoints, such as file information or relationships. The PDF export function, on the other hand, focuses on providing a human-readable representation of the same information. Finally, while the QR code export is not necessarily useful for the FAIR data package, it can be used to link digital objects in the form of records to their physical counterparts, which can be particularly useful for labelling samples or equipment in a laboratory. A notable addition to the export functionality is the ability to easily exclude certain exported record information, for example by using Kadi4Mat's GUI to select specific entries of the basic or generic metadata to be excluded, as shown in Figure 7.

For the FAIR data package, this function was primarily used to exclude the user information of the records, both those automatically captured and those contained in the generic metadata.

As already mentioned, Kadi4Mat offers the possibility to export the data and metadata of a record directly to a repository like Zenodo in order to publish it. While Kadi4Mat already automatically assigns globally persistent IDs to all resources created, selected resources that are to be published must be generally accessible and assigned a DOI for proper publication. As this export functionality is currently limited to individual records, as opposed to entire collections of records, and does not include additional export formats such as PDF, it was not used directly to publish the FAIR data package. Instead, as all relevant records were bundled into one collection, all data and metadata, both in user- and machine-readable form, of the records contained in this collection were automatically collected and bundled. This was again achieved via a small Python script [36] using the

kadi-apy library. The publication of the final FAIR data package on Zenodo was then done manually. While this process can be further improved into a more user-friendly interface, Kadi4Mat could also offer direct registration of DOIs in the future. However, this places additional demands on the long-term storage and preservation of data and metadata, especially with regard to the hardware on which the system runs.

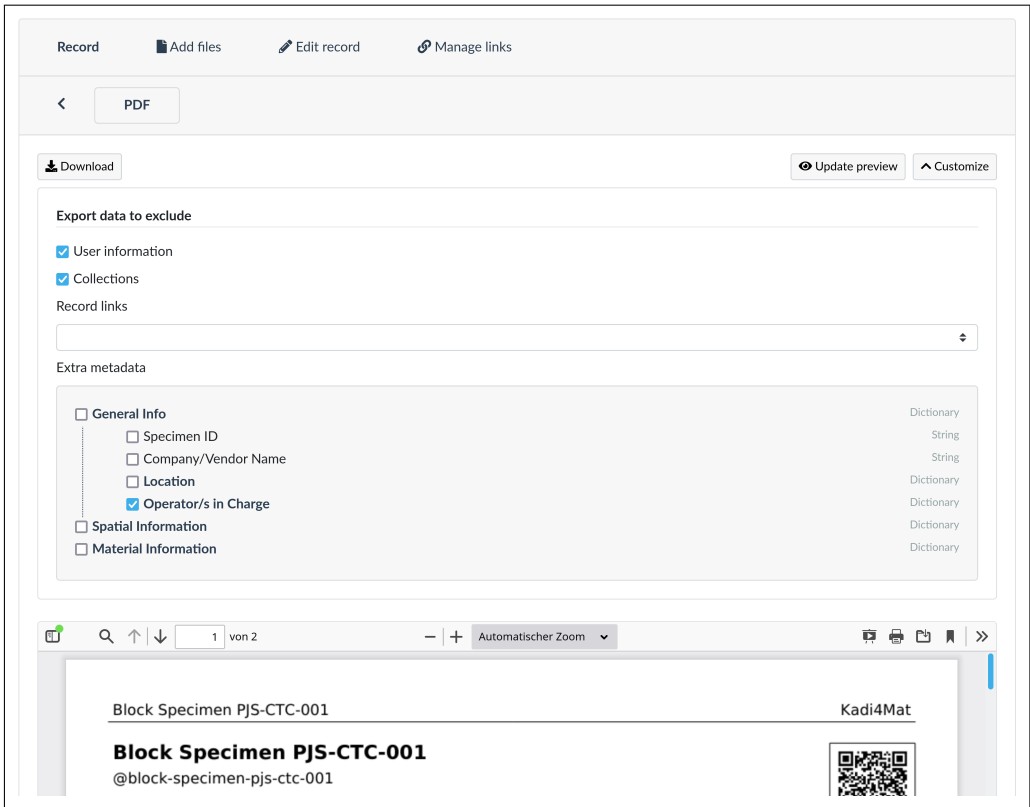

**Figure 7.** Screenshot showing the export mask of a record when exporting it as PDF. The record represents the test-ready counter body, as also shown in Figure 2. In this case, the general user information, the linked collections and a specific metadatum (*General Info > Operator/s in Charge*) are excluded in the generated PDF file.

## 4. Discussion

As part of the FAIR data package, presented in more detail in the sister publication [25], all data and metadata created and collected during the implementation of the showcase experiment are provided. In addition to human-readable information and visualisations necessary for reproducibility and ease of understanding, the data package contains machine-readable metadata that are also relevant for data science applications such as machine learning. To ensure the FAIRness of the published data package, the data object assessment metrics developed as part of the FAIRsFAIR project [8] were followed as closely as possible whenever they were applicable. These metrics, while still under development, provide guidance that is more detailed and practical than the FAIR principles themselves. Kadi4Mat already ensures that most metrics are satisfied, e.g., metadata contain provenance information about the origin of the data, while the use of the TriboDataFAIR ontology and Zenodo publication make sure that the data are formally described and publicly accessible. A detailed assessment of the metrics can be found in Table A1. Unfortunately, some metadata are not retrievable because the tribological samples and devices are partly provided by external vendors who often do not report all information.

As mentioned previously, Kadi4Mat was mainly used as an intermediary between the showcase experiment and the resulting data package as part of the FAIR data production pipeline. Working with the tribologists, the advantages of developing the system in parallel

with the work of the researchers who actually use it became apparent. In this way, all developed functions can be promptly tested in practice, while feedback can be directly discussed and implemented step by step. Tribology is an ideal candidate as it relies on customised and constantly changing workflows and therefore requires generic and flexible approaches, which is in line with Kadi4Mat's philosophy.

The showcase experiment conducted by the tribologists included not only the actual laboratory activities, but also a lot of preliminary work to digitise the experimental steps, including the collection of terms to create a well-defined vocabulary for an ontology. The flexibility of Kadi4Mat is particularly important for such top-down approaches, where researchers already have a comparatively clear structure for data management and workflows in mind. However, these approaches require not only a lot of work for the domain experts, but also interdisciplinary knowledge in fields other than tribology, such as ontology development. Moreover, the concepts, terms and ontologies developed need to be adapted over time, even when focusing on a specific type of experiment, as in this case. While there will always be initial hurdles before a data infrastructure can be used efficiently, even if it is only used in individual workgroups, extensive preparation up front should not be mandatory. Kadi4Mat must therefore also be able to support users and groups who have no prior knowledge of research data management.

For the most part, Kadi4Mat does not enforce a strict metadata schema, and digital objects such as records can be created without much prior knowledge or preparation, especially when the necessary guiding functionality already exists. In this way, researchers can get started immediately, and once common metadata are established, for example, in a single workgroup, templates can be used to facilitate the creation of consistent and comparable metadata and even metadata schemas where no suitable existing schemas are available. This can be further supported by improving the template functionality of Kadi4Mat, for example, by providing ways to enforce fixed metadata templates that more closely resemble an actual metadata schema, and by providing appropriate versioning capabilities to record the continuous evolution of such schemas. In addition, when changes are made to such templates, the metadata of the corresponding records in which they were used could be migrated semi-automatically, depending on the complexity of the changes, which is particularly important when requirements, and thus schemas, change frequently. By specifying relationships between the digital objects, using record links, and grouping multiple records into collections, a simple, ontology-like structure can be created from the bottom up. Although this structure, at least in its current state, is not a substitute for an actual ontology created with all the features that a format like OWL has to offer, it can serve as a basis for such or similar formal specifications. We believe that these bottom-up, practice-oriented approaches are essential to the eventual adoption of generic systems of this kind.

Finally, as with all digital infrastructures, proper administration of these systems is required. This refers not only to the initial installation, but also to ongoing maintenance, such as backups and keeping all necessary software packages up to date. While Kadi4Mat offers the possibility to set up private instances, for example at the level of individual workgroups, a hosted, out-of-the-box solution helps a lot with the short-term adoption of such systems and was also important in this work. In general, this choice depends on the requirements of data sovereignty, data sharing, security regulations, and hardware capabilities, and can vary greatly for each use case.

## 5. Conclusions

In this work, the research data infrastructure Kadi4Mat was used as part of an effort to digitise the data and metadata of a typical tribological experiment. Tribology was particularly well suited as an exemplary use case due to its heterogeneous and interdisciplinary nature. Working closely with the tribologists, the existing research data infrastructure was further adapted to meet their specific needs while remaining as generic as possible. In developing appropriate solutions for the joint project, both sides created various tools to



facilitate the manual entry and automatic collection of data and metadata. It is clear that as the developed solutions move closer to production, such as the use of Kadi4Mat's ELN functionality in everyday laboratory work, continuous efforts are required from all parties to further improve the tools and keep them up to date. Furthermore, the described use case represents only one example of the use of Kadi4Mat. Further best practice examples need to be developed and presented to show different valid approaches regarding how existing scientific workflows can be integrated independent of the research discipline and of the users' previous knowledge in research data management. This is especially important for such generic solutions, as too much flexibility can itself be an obstacle.

The future goal of Kadi4Mat is to support researchers throughout the entire research process, which in this case would mean being able to create a large part of the FAIR data package directly within the data infrastructure. One aspect that is particularly relevant in this regard is automated data acquisition to capture data and metadata as close to the source as possible. Kadi4Mat's HTTP API already provides an interface for automation, but it lacks the ease of use that a GUI can provide. One solution is the workflow functionality mentioned in Section 2, where the aim is to provide a graphical, user-friendly and optionally interactive interface.An integrated approach is under development to create, manage and execute such workflows, either locally via a separate package or directly via Kadi4Mat's web interface. Improved automation provides a more direct way to capture data provenance, which also includes the research software used for data analysis. Software representation in particular may require additional metadata, such as the software versions used, dependencies and deployment requirements [38], as well as versioned links to the respective evolving inputs and outputs. While this work has primarily tracked data collection, this is especially relevant for simulative or other data-intensive workflows, and appropriate solutions need to be developed as part of Kadi4Mat alongside the actual workflow functionality.

**Author Contributions:** Conceptualization, N.B., N.T.G., P.J.S., C.G. and M.S.; data Curation, N.T.G. and P.J.S.; investigation, N.T.G. and P.J.S.; software, N.B., E.S. and P.Z.; supervision, C.G. and M.S.; validation: N.B. and N.T.G.; visualization, N.B. and N.T.G.; writing—original draft preparation, N.B.; writing—review & editing, N.B., N.T.G., E.S., P.J.S., P.Z., C.G. and M.S. All authors have read and agreed to the published version of the manuscript.

**Funding:** This work was supported by the Federal Ministry of Education and Research (BMBF) in the project FestBatt (project number 03XP0174E), the German Research Foundation (DFG) in the projects POLiS (project number 390874152) and SuLMaSS (project number 391128822), the Ministry of Science Baden-Württemberg in the project MoMaF—Science Data Center, with funds from the state digitization strategy digital@bw (project number 57), the Federal Ministry of Education and Research (BMBF) and the Ministry of Science Baden-Württemberg as part of the Excellence Strategy of the Federal Government and the State Governments in the Kadi4X project, the Cx project, which is funded as part of the National High-Performance Computing (NHR) initiative, the European Research Council (ERC) under Grant No. 771237 (TriboKey) and the Alexander von Humboldt Foundation for awarding a postdoctoral fellowship to Nikolay Garabedian.

**Institutional Review Board Statement:** Not applicable.

**Informed Consent Statement:** Not applicable.

**Data Availability Statement:** The FAIR data package [24] of the tribological showcase experiment and the TriboDataFAIR ontology [30] are each available on Zenodo. The same applies to snapshots of the software used in this work, including Kadi4Mat [23], kadi-apy [27], SurfTheOwl [32] and the Python helper scripts [36].

**Acknowledgments:** We would like to thank Domenic Frank and Christof Ratz for their administrative support in maintaining the centralised Kadi4Mat instance used in this project.

**Conflicts of Interest:** The authors declare no conflict of interest.

## Appendix A

**Table A1.** Assessment on the FAIRness of the FAIR data package [24] published on Zenodo using the data object assessment metrics developed as part of the FAIRsFAIR project [8]. For each metric, specified by its identifier and name, the corresponding implementation is given in relation to the data package.

| Metric Identifier and Name | Implementation |
| --- | --- |
| FsF-F1-01D: Data is assigned a globally unique identifier. | Each individual record of the data package contains in its metadata a globally unique identifier provided by Kadi4Mat, both for the record as a whole and individual files, consisting of the base URL of the Kadi4Mat instance and a unique numeric ID. |
| FsF-F1-02D: Data is assigned a persistent identifier. | The data package as a whole is assigned a permanent identifier in the form of a DOI provided by Zenodo. |
| FsF-F2-01M: Metadata includes descriptive core elements (creator, title, data identifier, publisher, publication date, summary and keywords) to support data findability. | The listed domain-agnostic core metadata are defined via Zenodo for the data package as a whole. Individual records of the data package contain a subset of these metadata defined via Kadi4Mat (e.g., titles, identifiers, descriptions and tags), whereby metadata containing personal information about individual users are removed for data protection purposes. |
| FsF-F3-01M: Metadata includes the identifier of the data it describes. | Since the individual records in the data package serve as containers for related data and metadata, the latter always contain the identifier of the former within them (see also FsF-F1-01D). |
| FsF-F4-01M: Metadata is offered in such a way that it can be retrieved by machines. | Zenodo offers several APIs to retrieve and search the core metadata that are available, namely a REST API and an OAI-PMH [39] mechanism. Unfortunately, the technical metadata of the individual records in the data package cannot be retrieved in the same way (i.e., they cannot be searched directly), as they are part of the data package as a whole. Furthermore, Zenodo itself only records general metadata in a searchable way. The technical metadata nevertheless remain retrievable and their machine interoperability is given by the use of the JSON format. |
| FsF-A1-01M: Metadata contains access level and access conditions of the data. | The data package is publicly available on Zenodo (*Open Access*), which is reflected in the metadata provided by Zenodo. See also FsF-R1.1-01M for license information. |
| FsF-A1-02M: Metadata is accessible through a standardized communication protocol. | Since Zenodo offers a REST API (see also FsF-F4-01M), both the metadata of the data package and the package itself can be accessed via HTTP(S). |
| FsF-A1-03D: Data is accessible through a standardized communication protocol. | See FsF-A1-02M. |
| FsF-A2-01M: Metadata remains available, even if the data is no longer available. | Since a DOI is issued for the data package (see FsF-F1-02D), it is ensured that the metadata remain accessible. |

**Table A1.** *Cont.*

| Metric Identifier and Name | Implementation |
|---|---|
| FsF-I1-01M: Metadata is represented using a formal knowledge representation language. | The metadata contained in the data package are exported directly from Kadi4Mat, which does not directly use an RDF-based [40] representation due to its own specific requirements for describing the properties and relationships of resources. However, the TriboDataFAIR ontology [30], which is published alongside the data package and electronically referenced and versioned within Zenodo, is represented using the standardised knowledge representation language OWL. Since it forms the basis for the metadata contained in the data package, including the record relationships, the ontology directly serves as a formalised representation of the data package as a whole. There is also a JSON-LD [41] export function for some of the core metadata of the data package provided by Zenodo. |
| FsF-I1-02M: Metadata uses semantic resources. | See FsF-I1-01M. |
| FsF-I3-01M: Metadata includes links between the data and its related entities. | The related TriboDataFAIR ontology is linked to the data package as a whole through its DOI via Zenodo, which is reflected in the metadata provided by Zenodo. Where appropriate, individual records in the data package containing files include additional metadata that give further information about the purpose of these files. |
| FsF-R1-01MD: Metadata specifies the content of the data. | Each individual record of the data package contains rich metadata that are specified and exported via Kadi4Mat. Consequently, all data are described to the greatest extent possible by the associated metadata of the corresponding record. See also FsF-I3-01M. |
| FsF-R1.1-01M: Metadata includes license information under which data can be reused. | A Creative Commons Attribution 4.0 licence was chosen for the data package, specified at the level of the individual records (via Kadi4Mat) and for the data package as a whole (via Zenodo). This is reflected in the metadata contained in the data package and in the metadata provided by Zenodo, respectively. |
| FsF-R1.2-01M: Metadata includes provenance information about data creation or generation. | Since the relationships of all records representing the processes conducted in the showcase experiment are established via Kadi4Mat through record links, the provenance of the data is specified in the exported metadata as part of the data package. |
| FsF-R1.3-01M: Metadata follows a standard recommended by the target research community of the data. | As there are currently no established metadata schemas for experimental tribology, the metadata contained in the data package were captured via Kadi4Mat using various custom metadata templates, each describing a specific type of experimental process. However, this paper—and its sister publication [25]—attempt to ignite this process within the community. |
| FsF-R1.3-02D: Data is available in a file format recommended by the target research community. | Although there are currently no recommended file formats for tribological datasets, all data contained in the data package are provided in widely used and supported formats such as PDF, JSON or PNG, wherever possible. |

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
