# Peer review of "Managing FAIR Tribological Data Using Kadi4Mat"

_data, 2021_

Round 1

Reviewer 1 Report

This is nice paper -- overall good presentation that demonstrates a sound approach to making research data FAIR.

For readers not familiar with tribology it would be nice to have a bit more explanation about the experiment.  Cup grinding?  Counter body?  Base body?  Perhaps a photo of the experimental set-up could be included with the various elements labeled?

In section 3, 2nd paragraph, it's not clear what "analogue" form means with regard to the import of data.

Perhaps it is explained in the "sister publication" (not yet accepted?) as to how the FAIRness of the output data was assessed.  It is one thing to claim the data package is FAIR, another to have it independently verified.

Reviewer 2 Report

The authors present a very useful tool in the form of Kadi4mat, for storing tribology data as per FAIR guidelines. The overall theme and goals are well defined and the methods are to the point of the goals. I would like the authors to clarify the re-usability aspect of the generated data: How does it accommodate updated schemas? What does it take for the users to change the format in which data is stored, in other words, how flexible is it to update the schema? 

For example, if there is a dataset D1 which consists of information I1, then if the second dataset D2 consists of information I1 {union} I2, then how do you deal with it. And if you want to add another dataset D3 with information I1 \ I3, then how do you merge this information. In my opinion, this sense of backward/forward-compatibility is extremely critical for a dataset. Please elaborate / incorporate these into the framework.
